# Long-term exposure to outdoor air pollution and asthma in low-and middle-income countries: A systematic review protocol

**Achenyo Peace Abbah** [1] *, **Shanshan Xu** [1], **Ane Johannessen** [2]

**1** Department of Global Public Healthand Primary Care, Center for International Health, University of Bergen, Bergen, Norway, **2** Department of Global Public Health and Primary Care, University of Bergen, Bergen, Norway

* Achenyo.Abbah@student.uib.no

**Data Availability Statement:** No datasets were generated or analyzed during the current study. All relevant data from this study will be made available upon study completion.

## Abstract

### Background

Several epidemiological studies have examined the risk of asthma and respiratory diseases in association with long-term exposure to outdoor air pollution. However, little is known regarding the adverse effects of long-term exposure to outdoor air pollution on the development of these outcomes in low- and middle-income countries (LMICs). Our study aims to investigate the association between long-term exposure to outdoor air pollution and asthma and respiratory diseases in LMICs through a systematic review with meta-analysis.

### Methods

This systematic review and meta-analysis will follow the PRISMA (Preferred Reporting for Systematic Reviews and Meta-Analyses) checklist and flowchart guidelines. The inclusion criteria that will be used in our study are 1) Original research articles with full text in English; 2) Studies including adult humans; 3) Studies with long-term air pollution assessment in LMICs, air pollutants including nitrogen oxide ($NO_2$), sulfur oxide ($SO_2$), particulate matter ($PM_{2.5}$ and $PM_{10}$), carbon monoxide (CO) and ozone ($O_3$); 4) cohort and cross-sectional studies; 5) Studies reporting associations between air pollution and asthma and respiratory symptoms. A comprehensive search strategy will be used to identify studies published up till August 2022 and indexed in Embase, Medline, and Web of Science. Three reviewers will independently screen records retrieved from the database searches. Where there are enough studies with similar exposure and outcomes, we will calculate, and report pooled effect estimates using meta-analysis.

### Systematic review registration

PROSPERO CRD42022311326.

### Discussion

Findings from the health effects of long-term exposure to outdoor air pollution may be of importance for policymakers. This review will also identify any gaps in the current literature on this topic in LMICs and provide direction for future research.

**Funding:** The author(s) received no specific funding for this work.

**Competing interests:** The authors have declared that no competing interests exist.

## Introduction

Outdoor air pollution is a major menace to public health globally [1] that causes around 4.2 million deaths each year and inflicts a heavy morbidity burden on society [2, 3]. Sweileh and co-workers) [1] pointed out that almost 90% of deaths related to air pollution happen in low- and middle-income countries (LMICs) with almost 2 out of 3 happening in South-East Asia and Western Pacific regions. The World Health Organization (WHO) reports that 99% of the global population lives in regions where the recently launched WHO guideline limits on air pollution are exceeded [3, 4]. South and East Asian locations emerge as the most polluted globally. Bangladesh, China, India, and Pakistan share 49 of the 50 most polluted cities worldwide. This high air pollution rate in Asia can be related to expeditious urbanization and industrialization [1, 5]. In other areas such as the African Continent, there is no sufficient documentation on the magnitude of the attributable risk of outdoor air pollution [6, 7]. The problem of air pollution is particularly severe in countries with social disparities and a lack of sustainable management of the environment [8].

Duan and co-workers [9] showed that even though the levels of air pollution in high income countries have significantly decreased over the last 25 years, over the same period, air pollution levels are on the increase in LMICs especially China and India.

In recent years, there has been an increase in knowledge about health effects of long-term air pollution exposures, especially on asthma and respiratory symptoms. In several countries, the prevalence of asthma is between 1 and 8% of the population. A recent study has shown that 13% of global incidence of asthma in children can be attributable to traffic-related air pollution (TRAP) and TRAP affects the development of asthma also in adults [10]. In 2019, WHO reported that about 262 million persons had asthma which caused 461,000 deaths annually. Most asthma-related deaths happen in LMICs due to challenges of under-diagnosis and under-treatment) [11]. The WHO [11] further stressed the impact of asthma on normal daily living as it causes poor concentration, sleep disturbance, and tiredness during the day among persons not sufficiently treated for their asthma. Also, people who suffer from asthma and their families face the challenges of missing school and work, thus causing a substantial economic burden on their families and society at large.

Asthma is described as chronic inflammatory disorder of the airways associated with bronchial hyper-responsiveness, and reversible airflow limitation [5, 10, 12]. In other words, asthma occurs when the air passage in the lungs narrows because of inflammation and tightening of the muscles around the small airways. The main symptoms of asthma are wheeze, dyspnea, cough, tightness of the chest and shortness of breath.

Air pollution may induce or aggravate asthma. Pollutants in the atmosphere are linked with increased incidence, prevalence, hospitalizations, or worsening symptoms of asthma [12]. Tiotiu and co-workers [10] have further acknowledged that air pollution does not only worsen existing asthma but may cause new onset of asthma in previously healthy persons.

Even though both the air pollution burden and asthma disease burden are highest in LMICs, more systematic overviews from LMICs are scarce. One recent overview examined air pollution in LMICs in association with respiratory mortality and chronic obstructive pulmonary disease (COPD) [13], and one overview has examined air pollution in LMICs in association with asthma in children [14], but no overview currently exists covering air pollution and asthma in adults in LMICs. Such overviews could be important tools to improve public health by providing the basis for informed policy making and stimulating public health institutions and authorities to put more effective measures in place to reduce exposure to air pollutants. Thus, the main aim of this systematic review is to investigate the association between long-

term health effects of outdoor air pollution and asthma and respiratory symptoms among adults over 18 years old in LMICs.

## Materials and methods

The proposed systematic review and meta-analysis will be conducted following the PRISMA (Preferred Reporting Items for Systematic Reviews and Meta-Analyses) checklist (S1 Table) [15]. The study protocol was registered in advance in International Prospective Register of Systematic Reviews (PROSPERO- CRD42022311326).

### Objective

The main objective of this review is to investigate the association between long-term health effects of outdoor air pollution and asthma and respiratory symptoms among adults (over 18 years old) in LMICs.

### Review question

Does long-term exposure to outdoor air pollution increase the risks of asthma, and respiratory symptoms among adults (over 18 years old) in LMIC as compared to adults to those with relatively low levels of exposure to outdoor air pollution?

### Eligibility criteria

As pointed out by Schaefer and Mayers [16], documentation of clear criteria for inclusion and exclusion in any study is a major strength of the systematic review approach because it documents the reason why particular studies were selected as likely key studies and why other studies were excluded. These criteria are formed in accordance with the questions that are established during the problem formulation stage. In addition, eligibility criteria are conducted according to the Population (animal species inclusive), Exposure, Comparator, Outcomes, and Timing (PECOT) approach, study design, and date. Main exclusion criteria are unrelated studies, duplicates, full texts unavailability, or abstract-only papers while inclusion criteria entail studies on the target population, investigated exposure, or the comparison between two studied exposures. In a nutshell, the inclusion criteria should be articles that contain clear and sufficient information (both positive and negative) that answers the research question [17]. Based on this definition of eligibility criteria, we identified our inclusion and exclusion criteria for this study

### Inclusion criteria

- **Population:** Studies on human adult population on long-term exposure to outdoor air or gaseous pollutants (long-term defined as $\geq 1$ year in line with the 2021 WHO air quality guidelines [18] in LMICs.

- **Exposure:** Studies that reported on exposure to the outdoor air or gaseous pollutants nitrogen oxide ($NO_2$), sulphur oxide ($SO_2$), particulate matter ($PM_{10}$), particulate matter ($PM_{2.5}$), carbon monoxide (CO) and/or ozone ($O_3$).

- **Comparator:** Cohort studies that reported on exposure to relatively low levels of air or gaseous pollutants in the same population.

- **Outcomes:** Outcomes are asthma, and respiratory symptoms (such as wheeze, cough, and dyspnoea) that are not a result of biological agents.

- **Timing:** Studies conducted up to August 2022.

**Exclusion criteria.** Studies published in any other language besides English will not be included. Also, studies that are not available in full texts and studies conducted among participants less than 18 years old will not be considered.

## Information sources

A significant component of the systematic review process is sufficient searching of scientific and relevant literature, hence, the suggestion of Schaefer and Mayers) [16] that search terms should be carefully chosen to adequately narrow down the search results to produce rich information will be adhered to in this review.

As advised by Cheung and Vijayakumar [19] it is wise to search more than one database in order to reduce selection bias, thus, studies published in English up to August 2022 that matched the PECOT question will be searched systematically in Embase (Ovid), Medline (Ovid), and Web of Science (Core collection). The aforementioned databases were selected to avoid too many duplications of studies from similar databases such as PubMed, Embase, and the Cochrane Library. Furthermore, a librarian from the faculty of medicine at the university will guide the search.

## Search strategy

A pilot search of Embase (Ovid) was carried out to identify studies on the topic. The text words in the titles and abstracts of relevant studies and the index terms used to define the articles were used to develop a full search strategy (S2 Table). The reference lists of key full text articles included in the review will be screened to find additional studies. This review will conduct searches for relevant literature on the identified databases through a combination of free text and indexed terms such as Medical Subject Headings (MeSH) terms and will be combined using Boolean operators.

## Data management

The search results will be compiled in a reference manager program using EndNote and duplicates will be removed. The records will further be exported to Rayyan software [20] where the screening of all records will be done.

## Selection process

For a thorough review, three reviewers will independently screen all records retrieved from the database searches using Rayyan, a software for systematic reviews by screening the titles, and abstracts using screening question according to the inclusion and exclusion criteria. Full text articles will then be assessed by the same reviewers. Any disagreements between the three independent reviewers will be resolved through group discussion and voting or by contacting the author if further information is required. Reasons for exclusion of the articles will be recorded.

A PRISMA flowchart as shown in (S1 Fig) representing the selection process and numbers of the selected articles, the numbers of the articles initially identified, the numbers of articles excluded before and after screening based on titles and abstracts, eligible articles did not meet inclusion criteria, and the primary reasons for exclusion [15] will be presented.

## Data collection process

Data from full texts will be extracted and screened by exporting the results to Excel form designed by the reviewers. The extraction characteristics of the included articles will be: name

of authors; year, journal/issue of publication; study location; study design; sample size of the study; demographic characteristics of the study population; pollutants; outcomes; statistical methods; effect estimates; confounders in the statistical model [21, 22]. In accordance with good practice, a pilot testing of the Excel form will be done on a sample of included studies to ensure that all relevant information is captured [23]. The three reviewers will compare and discuss the accuracy and completeness of the data extracted. If during the extraction process some data is missing, unclear or incomplete, inquiries will be sent to the authors.

## Risk of bias assessment

Three reviewers will independently assess the risk of bias in included by using the Risk of Bias In Non-randomized Studies-of Exposure (ROBINS-E) tool. The ROBINS-E tool developed by the ROBINS-E Development Group led by Higgins and co-workers [24] provides an orderly way to assess the risk of bias (RoB) in observational epidemiological studies. It includes seven domains of bias: confounding, exposure classification, participant selection, departure from intended exposure, missing data, and outcome measurement. Each domain is addressed using a series of signaling questions with the purpose of collecting significant information on the study and analysis being evaluated. In addition, three judgements are done after the important signaling questions have been answered, then, an overall judgement is carried out for each of these considerations [24]. This tool was used by Park and co-workers [25] in a very simplified approach, thus making it adoptable for other researchers like us.

A pilot quality assessment will be conducted on a few selected included studies. It is anticipated that the quality assessment of studies could involve a certain extent of subjective judgment, thus, any differences in opinion will be resolved through discussion. The quality assessment for individual included studies will be qualitatively summarized as part of the summary of the findings table.

## Analysis

**Descriptive analysis.**   We will conduct a narrative synthesis of the findings from the included studies. We will structure the narrative synthesis by describing the studies according to the study design; characteristics of the target population (e. g age, sex, socioeconomic status, educational level etc.); the type of air pollutants; the type of respiratory health outcomes.

**Statistical analysis.**   If there are enough studies with similar exposure and outcomes, we will pool the results using meta-analysis in the STATA software. As described by Cheung and Vijayakumar [19] and Lee [26], two statistical models are used for a meta-analysis given that a meta-analysis merges the effect sizes of the included studies by weighting the data in accord with the diverse amounts of data in each study. On one hand, the fixed effect model infers that all the studies in the meta-analysis have one true effect size and the observed variation amongst studies is due to sampling errors or chance. The fixed effect model evaluates only intra-study sampling errors, that is, intra-study variation. On the other hand, the random effect model assumes that various studies display considerate diversification, and the true effect size might range between studies. It also evaluates both intra-study sampling errors and inter-study variance, that is, between-study variation.

With the understanding of which model to use as described by Lee above, the DerSimonian and Laird random-effects methods for meta-analysis might be employed. This is in line with other systematic reviews and meta-analyses related to this topic of interest [21, 25, 27–29]. Furthermore, DerSimonian and Laird random effects model has been known to be the simplest and most widely used method for fitting the random effects model for meta-analysis [30]. Heterogeneity among studies will be assessed using both the $\chi^2$ test and $I^2$ statistics. According to

the Cochrane Handbook [31], we will consider an $I^2$ value over 50% to indicate substantial heterogeneity. We will assess publication bias by using funnel plots and Egger's linear regression.

## Assessment of certainty of evidence across studies

For each pollutant exposure and outcome, the certainty of evidence (CoE) will be judged by adapting the Grading of Recommendations Assessment, Development and Evaluation (GRADE) approach. The GRADE domains consist of risk of bias, directness of information, precision of an estimate, consistency of estimates across studies, risk of bias related to selective reporting, strength of the association, presence of a dose-response gradient, and the presence of plausible residual confounding that can increase confidence in estimated effects) [32]. The basis of the GRADE domains assessment will be from the results of the risk RoB assessment, heterogeneity, sensitivity, and publication bias analyses [21, 33]. The overall rating of certainty of evidence as described by [28] are as follows;

- High: means there is unlikely change in the effect estimate given further studies.

- Moderate: a certain likelihood in change of the effect estimate given further studies.

- Low: further studies are very likely to cause a change in the effect estimate.

- Very low: high uncertainty in the effect estimate.

## Discussion

This systematic review protocol was precisely developed to increase the knowledge and awareness on deleterious air or gaseous pollutants that cause obstructive respiratory diseases such as asthma and respiratory symptoms to support the drive for future research. There is growing evidence of the positive association between some air or gaseous pollutants and the development of respiratory diseases in high-income countries. However, very few studies have been conducted in the low-and middle-income countries on the significant association between air or gaseous pollutants and respiratory diseases. Hence, we aim to investigate the association between long-term health effects of outdoor air pollution and asthma and respiratory symptoms among adults (over 18 years old) in LMICs. This evidence will provide institutional bodies with a better prospect to regulate and formulate measures on air pollution to prevent unfavorable health outcomes in these countries.

## Conclusions

The findings from this review will contribute to the growing body of knowledge of the health effects of outdoor air pollution and may hopefully be used to inform policymaking in LMICs-contributing to improving public health in these areas.

## Supporting information

**S1 Table. Completed PRISMA-P checklist.**
(DOCX)

**S2 Table. Search strategy.**
(DOCX)

**S1 Fig. PRISMA flowchart of the study identification and selection process.**
(DOCX)

## Acknowledgments

We would like to acknowledge the contribution of Elisabeth Ebner for her assistance with the electronic pilot search strategy.

## Author Contributions

**Conceptualization:** Achenyo Peace Abbah.

**Formal analysis:** Achenyo Peace Abbah, Shanshan Xu.

**Methodology:** Achenyo Peace Abbah, Shanshan Xu, Ane Johannessen.

**Supervision:** Shanshan Xu, Ane Johannessen.

**Writing – original draft:** Achenyo Peace Abbah.

**Writing – review & editing:** Achenyo Peace Abbah, Shanshan Xu, Ane Johannessen.

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
