## [Decision Letter · Decision Letter 0]

3 Apr 2023

PONE-D-23-07292Long-term exposure to outdoor air pollution and asthma in low-and middle-income countries: a systematic review protocolPLOS ONE

Dear Dr. Abbah,

Thank you for submitting your manuscript to PLOS ONE. After careful consideration, we feel that it has merit but does not fully meet PLOS ONE’s publication criteria as it currently stands. Therefore, we invite you to submit a revised version of the manuscript that addresses the points raised during the review process.

For completeness please ensure you provide response to the following comments raised by the reviewers: 

We look forward to receiving your revised manuscript.

Kind regards,

Haruna Musa Moda

Academic Editor

PLOS ONE

Journal Requirements:

2. We note that the original protocol file you uploaded contains a confidentiality notice indicating that the protocol may not be shared publicly or be published. Please note, however, that the PLOS Editorial Policy requires that the original protocol be published alongside your manuscript in the event of acceptance. Please note that should your paper be accepted, all content including the protocol will be published under the Creative Commons Attribution (CC BY) 4.0 license, which means that it will be freely available online, and any third party is permitted to access, download, copy, distribute, and use these materials in any way, even commercially, with proper attribution.

Therefore, we ask that you please seek permission from the study sponsor or body imposing the restriction on sharing this document to publish this protocol under CC BY 4.0 if your work is accepted. We kindly ask that you upload a formal statement signed by an institutional representative clarifying whether you will be able to comply with this policy. Additionally, please upload a clean copy of the protocol with the confidentiality notice (and any copyrighted institutional logos or signatures) removed.

- https://www.frontiersin.org/articles/10.3389/fped.2021.827507/full

In your revision ensure you cite all your sources (including your own works), and quote or rephrase any duplicated text outside the methods section. Further consideration is dependent on these concerns being addressed

Reviewers' comments:

Reviewer's Responses to Questions

**Comments to the Author**

1. Does the manuscript provide a valid rationale for the proposed study, with clearly identified and justified research questions?

Reviewer #1: Yes

Reviewer #2: Partly

2. Is the protocol technically sound and planned in a manner that will lead to a meaningful outcome and allow testing the stated hypotheses?

Reviewer #1: Yes

Reviewer #2: Yes

3. Is the methodology feasible and described in sufficient detail to allow the work to be replicable?

Reviewer #1: Yes

Reviewer #2: Yes

4. Have the authors described where all data underlying the findings will be made available when the study is complete?

Reviewer #1: Yes

Reviewer #2: Yes

5. Is the manuscript presented in an intelligible fashion and written in standard English?

Reviewer #1: Yes

Reviewer #2: Yes

6. Review Comments to the Author

You may also provide optional suggestions and comments to authors that they might find helpful in planning their study.

Reviewer #1: Make sure to accentuate the findings of the systematic review that relates to /answer the review question you stated in your protocol.

Reviewer #2: Detailed Review

The current manuscript details on a study protocol for carrying out a systematic review on the long-term exposure to outdoor air pollution and its links to asthma prevalence in low-and middle-income countries (LMIC).

General Comments

The proposed review could help in understanding the real effects of urbanization in LMIC countries and its long lasting effects on public health in these countries. Hence, it’s a needed effort in the community.

G1 - The English language in the manuscript needs to be more polished

e.g. 1 - Refer lines 61 to 63. The wordings are not right

e.g. 2 - Instead of using Dual et al., Orlenno et al., and so forth in the paper, rewrite these sentences to bring in more uniformity for the reader.

G2 - Certain sections should be rewritten more concisely

E.g. – Section on “Assessment of certainty of evidence across studies”

In here, once the author expands the GRADE approach and cites appropriately, there is no need to write in detail and define it. One of the important purpose of citing/quoting references is also to refer the reader to sources which can provide more information on the context being discussed and to avoid detailing the concepts or procedures again.

Major Comments

MAJ 1 - Lines 84-85 – Mentions systematic overview from LMICs is lacking. This definitely is an understatement as recent systematic reviews and meta-analysis are available on this topic (https://doi.org/10.1016/j.envres.2022.114604;
https://doi.org/10.1016/j.atmosenv.2021.118422).

Hence, the introduction needs to be rewritten with better emphasize on the need for the current proposed review and how it is different/unique from other existing reviews on the topic.

MAJ 2 - Lines 249-251 - The expected outcomes of the review are over stated. How are the authors linking a review of this kind to directly inform policymaking?

MAJ 3 - In their inclusion criteria, the authors mention “Long term” and “≥1 year”. Typically, at least a period covering more than one annual cycle or with repetitive annual seasonal cycles are referred to as Long term. Hence, use a different jargon or increase the exposure duration to “≥2 years” if feasible.

Minor Comments

MIN 1 - Lines 36 to 42 – The historical statements given here are of little importance to the context. The authors should rewrite these in a better way leading the reader to the topic.

MIN 2 - Lines 125-126 – “Outcomes that include asthma, respiratory symptoms” – Such outcomes can also result from biological agents. So care should be taken while screening articles and this needs to be explicitly mentioned in the inclusion/exclusion criteria’s

MIN 3 - Introduce expanded forms of abbreviations before start using them (e.g. RoB)

MIN 4 - Please check for appropriate use of subscripts and post scripts throughout the manuscript including the reference section.

MIN 5 - The authors assume 18 years to be the universal adult age and this is not true even within some LMIC countries. So it’s better to write it explicitly as 18 years through the manuscript in place of saying “adults”

7. PLOS authors have the option to publish the peer review history of their article (what does this mean?). If published, this will include your full peer review and any attached files.

Reviewer #1: No

Reviewer #2: **Yes: **Ravi Rangarajan

---

## [Author Response · Author response to Decision Letter 0]

13 Jun 2023

Reviewer #1: 

Comment 1: Make sure to accentuate the findings of the systematic review that relates to /answers the review question you stated in your protocol.

Response: Thank you for pointing this out, we will indeed accentuate the findings relating to the review question in our systematic review and keep this protocol paper vivid in mind when performing the systematic review.

Reviewer #2: Detailed Review

The current manuscript details a study protocol for carrying out a systematic review of long-term exposure to outdoor air pollution and its links to asthma prevalence in low-and middle-income countries (LMIC).

General comment 1: The English language in the manuscript needs to be more polished

e.g., 1 - Refer to lines 61 to 63. The wordings are not right

e.g., 2 - Instead of using Dual et al., Orlenno et al., and so forth in the paper, rewrite these sentences to bring in more uniformity for the reader.

Response: Thank you for pointing this out. We agree with this comment and have gone through the manuscript rephrasing wordings in better English where needed to improve the reader’s understanding. 

e.g., 1: We have corrected the wording. The change can for example be found on page 3, paragraph 3, and lines 63-64. 

e.g., 2- We have rewritten the sentences by incorporating the names of the authors using in-text Vancouver referencing style all through the manuscript (writing, for example, Duan and co-workers instead of Duan et al. on page 3, paragraph 2, and lines 60).

General comment 2: Certain sections should be rewritten more concisely

E.g. – Section on “Assessment of certainty of evidence across studies”

Here, once the author expands the GRADE approach and cites appropriately, there is no need to write in detail and define it. One of the important purposes of citing/quoting references is to refer the reader to sources that can provide more information on the context being discussed and avoid detailing the concepts or procedures again.

Response: Agree. We have, accordingly, revised this sentence and we have deleted the detailed description. We have made the changes on page 11, lines 235-237.

Major comment 1: Lines 84-85 – Mentions systematic overview from LMICs is lacking. This definitely is an understatement as recent systematic reviews and meta-analyses are available on this topic (https://doi.org/10.1016/j.envres.2022.114604;
https://doi.org/10.1016/j.atmosenv.2021.118422).

Hence, the introduction needs to be rewritten with a better emphasis on the need for the current proposed review and how it is different/unique from other existing reviews on the topic.

Response: We thank the reviewers for the two review papers’ recommendations. Our statement about how systematic overviews from LMICs are lacking refers to overviews on air pollution and asthma in adults. We agree this should be more clearly specified and have rewritten the Introduction accordingly. We have also included the two suggested references (page 4, lines 89-93).

Major comment 2: Lines 249-251 - The expected outcomes of the review are overstated. How are the authors linking a review of this kind to directly inform policymaking?

Response: we agree the expected outcomes may have been worded a bit too ambitiously, and we have revised the manuscript accordingly. Although systematic review papers such as our planned paper are indeed suitable to inform policymaking- through summarizing large amounts of information, identifying positive and negative effects of various exposures, and identifying gaps in medical research- we do not have a direct link with the policymakers in LMICs and can therefore not assume that our paper will directly inform policymaking. We have kept the potential for policymaking in the manuscript but have acknowledged that we cannot be certain that our paper will be used for this purpose (page 5, lines 93-98, page 12, lines 258-260, lines 262-264).

Major comment 3: In their inclusion criteria, the authors mention “Long term” and “≥1 year”. Typically, at least a period covering more than one annual cycle or with repetitive annual seasonal cycles are referred to as long term. Hence, use a different jargon or increase the exposure duration to “≥2 years” if feasible.

Response: Although we agree that more than one annual cycle is ideal for looking at long-term exposures, the purpose of this planned systematic review is to gather an overview of all long-term air pollution exposure papers focusing on asthma in adults in LMICs. To not miss out on any papers, we have chosen the definition of long-term exposure from the 2021 WHO Global air quality guidelines. In these guidelines, long-term exposure is defined as “a mean of one or several years” while short-term exposure is “measures over minutes to days”. We have specified the reason for our definition of long-term exposure in the revised manuscript (page 6, lines 128-129).

Minor comment 1: Lines 36 to 42 – The historical statements given here are of little importance to the context. The authors should rewrite these in a better way leading the reader to the topic.

Response: Thank you for pointing this out. We have, accordingly, changed the first paragraph in the introduction section for better understanding. This can be found on pages 2-3, lines 38-46. 

Minor comment 2: Lines 125-126 – “Outcomes that include asthma, respiratory symptoms” – Such outcomes can also result from biological agents. So care should be taken while screening articles and this needs to be explicitly mentioned in the inclusion/exclusion criteria

Response: Agree. Health outcomes that were identified in this systematic review are now explicitly mentioned in the inclusion/exclusion criteria. This can be found on page 6, lines 135-136.

Minor comment 3: Introduce expanded forms of abbreviations before start using them (e.g., RoB)

Response: We agree with this and have incorporated your suggestion throughout the manuscript. The expanded form of RoB can be found on page 11, and line 242.

Minor comment 4: Please check for the appropriate use of subscripts and postscripts throughout the manuscript including the reference section.

Response: We agree with this and have incorporated your suggestion throughout the manuscript. These changes can for example be found on page 13, lines 325, 350-355.

Minor comment 5: The authors assume 18 years to be the universal adult age, and this is not true even within some LMIC countries. So, it’s better to write it explicitly as 18 years through the manuscript in place of saying “adults”

Response: We thank the reviewers for pointing this out. We have explicitly stated that we mean 18 years or above when referring to adults throughout the revised manuscript. These changes can for example be found on page 5, lines 106-107.

---

## [Decision Letter · Decision Letter 1]

2 Jul 2023

Long-term exposure to outdoor air pollution and asthma in low-and middle-income countries: a systematic review protocol

PONE-D-23-07292R1

Dear Dr. Abbah,

We’re pleased to inform you that your manuscript has been judged scientifically suitable for publication and will be formally accepted for publication once it meets all outstanding technical requirements.

Kind regards,

Haruna Musa Moda

Academic Editor

PLOS ONE

Additional Editor Comments (optional):

Reviewers' comments:

Reviewer's Responses to Questions

**Comments to the Author**

1. Does the manuscript provide a valid rationale for the proposed study, with clearly identified and justified research questions?

Reviewer #1: Yes

Reviewer #2: Yes

2. Is the protocol technically sound and planned in a manner that will lead to a meaningful outcome and allow testing the stated hypotheses?

Reviewer #1: Yes

Reviewer #2: Yes

3. Is the methodology feasible and described in sufficient detail to allow the work to be replicable?

Reviewer #1: Yes

Reviewer #2: Yes

4. Have the authors described where all data underlying the findings will be made available when the study is complete?

Reviewer #1: No

Reviewer #2: Yes

5. Is the manuscript presented in an intelligible fashion and written in standard English?

Reviewer #1: Yes

Reviewer #2: Yes

6. Review Comments to the Author

You may also provide optional suggestions and comments to authors that they might find helpful in planning their study.

Reviewer #1: There are a lot of improvements in this current submission.

I would like to propose a suggestion regarding the repetition of the phrase "over 18 years old" in the Review question. Please consider rephrasing it for conciseness, for example:

Does long-term exposure to outdoor air pollution increase the risks of asthma and respiratory

symptoms among adults (over 18 years old) in LMIC as compared to those

with relatively low levels of exposure to outdoor air pollution?

Reviewer #2: The authors have addressed all the concerns raised.

The manuscript is publishable in the current format.

7. PLOS authors have the option to publish the peer review history of their article (what does this mean?). If published, this will include your full peer review and any attached files.

Reviewer #1: No

Reviewer #2: **Yes: **Ravi Rangarajan

---

## [Editor Report · Acceptance letter]

11 Jul 2023

PONE-D-23-07292R1 

Long-term exposure to outdoor air pollution and asthma in low-and middle-income countries: a systematic review protocol 

Dear Dr. Abbah:

I'm pleased to inform you that your manuscript has been deemed suitable for publication in PLOS ONE. Congratulations! Your manuscript is now with our production department. 

Kind regards, 

on behalf of

Dr. Haruna Musa Moda 

Academic Editor

PLOS ONE